# Patient-Reported Outcomes, Health-Related Quality of Life, and Clinical Outcomes for Urothelial Cancer Patients Receiving Chemo- or Immunotherapy: A Real-Life Experience

**DOI:** 10.3390/jcm10091852

**Published:** 2021-04-24

**Authors:** Gry Assam Taarnhøj, Henriette Lindberg, Christoffer Johansen, Helle Pappot

**Affiliations:** 1Department of Oncology, University Hospital of Copenhagen, Rigshospitalet, 2100 Copenhagen, Denmark; christoffer.johansen@regionh.dk (C.J.); helle.pappot@regionh.dk (H.P.); 2Department of Oncology, University Hospital of Copenhagen, Herlev Hospital, 2730 Herlev, Denmark; henriette.lindberg@regionh.dk

**Keywords:** urothelial cell carcinoma, bladder cancer, quality of life, patient-reported outcomes, side effects, chemotherapy, immunotherapy

## Abstract

Patients with urothelial cell carcinoma (UCC) often have comorbidities, which cause trouble for the completion of oncological treatment, and little is known about their quality of life (QoL). The aim of the present study was to obtain and describe patient-reported outcomes (PRO) and QoL data from UCC patients in the treatment for locally advanced muscle-invasive or metastatic UCC. A total of 79 patients with UCC completed four questionnaires (EORTC QLQ-C30, QLQ-BLM30, HADS, and select PRO-CTCAE™ questions) once weekly during their treatment. From those, 26 patients (33%) underwent neoadjuvant treatment for local disease while 53 patients (67%) were treated for metastatic disease. Of all patients, 54% did not complete the planned treatment due to progression, nephrotoxicity, death, or intolerable symptoms during treatment. The five most prevalent PRO-CTCAE grade ≥ 2 symptoms were frequent urination (37%), fatigue (35%), pain (31%), dry mouth (23%), and swelling of the arms or legs (23%). The baseline mean overall QoL was 61 (±SD 24) for all patients (neoadjuvant (73, ±SD 19) and metastatic (54, ±SD 24)) and remained stable over the course of treatment for both groups. A stable overall QoL was observed for the patients in this study. More than half of the patients did not, however, complete the planned treatment. Further supportive care is warranted for bladder cancer patients.

## 1. Introduction

Patients with urothelial cell carcinoma (UCC) often have several comorbidities including renal and cardiopulmonary impairment [1,2]. These comorbidities and their treatment provide a major source of complications when introducing systemic treatment for cancer and may cause trouble for treatment adherence due to increased toxicity [3,4]. This in turn may diminish the effect of chemotherapy, especially for patients receiving neoadjuvant treatment, for whom the overall effect of chemotherapy is modest, irrespective of other existing conditions at treatment initiation [5,6,7]. With the limited survival effect in mind, quality of life (QoL) should be an important tool in monitoring patients during therapy [1,2,6]. It therefore seems relevant to register QoL during treatment for these patients. However, no QoL data from UCC patients receiving standard chemotherapy as neoadjuvant treatment or for metastatic disease exist [8]. The current knowledge on the QoL of these patients is based on data from clinical trials [8]. However, it is well know, that such data may not be representative of patients with UCC receiving standard treatment, as patients participating in clinical trials often are more fit and have less comorbidities than patients receiving standard treatment outside of clinical trials [9,10]. In addition, the degree of adherence to treatment and the frequency of hospitalizations due to complications during treatment is largely unknown in this population [6]. Data on hospitalizations during treatment for lung cancer patients, with a similar comorbidity profile as UCC patients, indicate that the rate of hospitalization is high (23–56%) [11,12,13]. The novel focus on obtaining patient-reported outcomes (PROs) for cancer patients during treatment has been accompanied by reports of better QoL and, in a few studies, improved survival, most likely due to early management of symptoms during oncological treatment [12,14,15]. One may therefore assume that the early detection of symptoms with PROs potentially reduces hospitalizations, leads to fewer dose-reductions, and improves adherence to treatment [12].

The aim of this study is to systematically report patient-experienced symptomatology, health-related quality of life (HRQOL), and clinical outcomes in terms of treatment adherence and hospitalizations in a clinical prospective study of previously understudied UCC patients receiving standard chemotherapy as a neoadjuvant treatment, or chemo- or immunotherapy for metastatic disease.

## 2. Materials and Methods

### 2.1. Population

From 1 August 2017, all patients referred for oncological treatment with a diagnosis of muscle-invasive or metastatic bladder cancer at the Department of Oncology at the Copenhagen University Hospitals Rigshospitalet and Herlev Hospital, Denmark, fulfilling the following criteria were asked to participate. In this study, bladder cancer comprised urothelial cancer of the renal pelvis, ureter, bladder, or urethra. No patients with non-muscle-invasive disease were included in the sample. The following are the inclusion criteria:Age ≥ 18 years.Initiating chemotherapy as a neoadjuvant treatment, or chemo- or immunotherapy as a standard therapy (outside clinical trials) for metastatic disease.Able to read Danish.No serious cognitive impairment.

Besides the collection of PROs and QoL, this study was planned to provide information on clinical outcomes for a randomized study in the bladder cancer population (www.clinicaltrials.gov, accessed on 22 April 2021, NCT03584659), and as such, the study sample was collected according to the recommendations for pilot studies [16,17]. Enrolment was planned until a minimum of 20 patients for each treatment modality (chemo- and immunotherapy) were included. As immunotherapy as a standard treatment in Denmark was not available at the time of study initiation, enrolment started when patients received chemotherapy and continued until 20 patients receiving immunotherapy were enrolled. All treatments were given once every three weeks.

### 2.2. Study Design

The study design was a single-arm prospective descriptive study. The study was carried out prospectively for all patients who signed the informed consent form. There was no real-time feedback to patients when completing the questionnaires. However, for patients completing PROs electronically, the self-reported symptoms were available to the clinician at each clinical visit; see below. The study was conducted in accordance with the Declaration of Helsinki, and the protocol was approved by the Danish National Data Protection Agency (file no., 2012-58-0004). In Denmark, no ethical approval is needed for scientific studies assembling questionnaire data only.

### 2.3. Questionnaires

Participants were asked to complete the following questionnaires at baseline on the day of treatment initiation and once a week whilst receiving treatment: EORTC QLQ-C30 and QLQ-BLM30, HADS, and select PRO-CTCAE™ questions. The data were collected on paper or by electronic reporting through the Internet using a software system, Ambuflex [18], integrated into an existing clinical system. As the purpose of this study was the collection of PROs and clinical outcomes, the physicians were informed of the completed electronic questionnaires at clinical visits and were informed to handle the specific reported symptoms as preferred. No compulsory workflow in response to the questionnaires was implemented. However, a five-step implementation strategy was performed before initiation of the study in order to secure the compliance of the physicians and incorporation of the PROs into daily clinical practice for the participants of this study. The five steps comprised: (1) collective teaching for physicians and nurses; (2) instructions on how to access the ePRO system reviewed one-to-one for all physicians with outpatient duties in both uro-oncological outpatient clinics; (3) laminated instructions cards present and visible in all consultation rooms; (4) for every patient visit, every third week, the ambulatory programs marked with the study name (iBLAD) and a reminder to check the patient’s symptoms in the Ambuflex software; and (5) everyday telephone hot-line service assistance provided by the study investigator (G.A.T.), if needed. Further details regarding these procedures can be reviewed in a separate publication [19].

The baseline questionnaire was completed on the day initiating treatment, which was on the day of study inclusion. Patients completing the questionnaires on paper were asked to complete the questionnaires one week apart and to write the date of completion on the front page of the questionnaires. Patients completing the questionnaires electronically were sent an e-mail once per week with a link to the questionnaire and were prompted with e-mail reminders the following two days after the initial e-mail. Patients ceased symptom reporting after a maximum of six treatment cycles or when terminating treatment (if earlier), for whatever reason.

#### 2.3.1. EORTC

The European Organisation for Research and Treatment of Cancer (EORTC) QLQ-C30 was initially developed in 1987 (as QLQ-C36) and later revised to the version used today (v. 3.0) in 1997. The QLQ-C30 is a core questionnaire with 30 questions for assessing QoL in cancer patients and comprises nine multi-item scales, five functional scales, three symptom scales, and a global health scale [20]. It has been validated and translated into more than 100 languages, including Danish [21]. The BC specific module for muscle-invasive disease, QLQ-BLM30, can be used as a supplement to the QLQ-C30. It consists of 30 questions, divided into items concerning, e.g., urinary, bowel, and sexual functions [22]. The QLQ-C30 and QLQ-BLM30 both use a 0–100 scale, but whilst a higher global QoL score indicates better QoL, higher symptom scores for, e.g., urinary symptoms, pain, and fatigue, indicate increased impairment.

#### 2.3.2. HADS

The Hospital Anxiety Depression Scale (HADS) is a brief but commonly used scale initially developed in 1983 and used widely since [23,24]. The scale consists of 14 questions, divided into two subscales for anxiety and depression. The patients rate items on a 4-point scale. A score of 8–10 on either scale indicates a mild level of symptoms, whilst a score of 11 or more indicates clinical depression or moderate to severe anxiety [24,25]. HADS was included in this study to determine the prevalence of anxiety or depression in the BC population receiving chemotherapy.

#### 2.3.3. PRO-CTCAE

To address the growing need for a patient-led supplement to the clinician-based CTCAE reporting in clinical trials, the National Cancer Institute developed the Patient-Reported Outcomes version of the Common Terminology Criteria for Adverse Events [26]. The PRO-CTCAE library comprises 124 items representing 78 symptomatic toxicities, from which the relevant items are chosen depending on the study population. The PRO-CTCAE was translated into Danish in 2016 [27]. Responses are provided on a 5-point Likert scale (0–4), and the standard recall period is “the past 7 days”.

From the full PRO-CTCAE library, a subset of PRO-CTCAE questions for this study were chosen specifically for this population through a mixed-methods process of journal audits, patient interviews, EMA and FDA document reviews, and review of the randomized trials leading to approval of the applied immunotherapies. The process is described in detail elsewhere [28]. The final PRO-CTCAE questionnaire used in this study comprised 45 symptoms explored by a total of 84 PRO-CTCAE questions. The PRO-CTCAE questions were included in this study to determine the prevalence and severity of symptoms in this understudied population.

### 2.4. Outcomes

Based on the collected data, the most frequent and severe symptoms for this population along with the development of the patients’ overall HRQOL are reported. The clinical outcomes for this patient group in terms of rate of completion, reasons for discontinuation, and rate of hospital admissions (if there are multiple for the same patient, only the first hospital admission is be reported) is also be reported in this study.

### 2.5. Statistics

The data were descriptively analyzed in the statistical software SPSS Statistics version 25.0. Student’s T-tests and linear regression models were performed to detect differences in QoL, summarized PRO scores, and clinical outcomes between groups of patients who experienced early treatment cessation or hospitalization vs. patients who did not.

## 3. Results

### 3.1. Inclusion

From 1 August 2017 to 21 September 2018, a total of 95 patients fulfilling the inclusion criteria were approached before starting chemo- or immunotherapy. Three patients (3%) declined treatment altogether, and three patients (3%) were missed at treatment initiation. A further ten patients (11%) declined participation, leaving 79 patients who signed the written informed consent. Data collection was completed and finalized on 21 January 2019.

### 3.2. Clinical Data

The demographical data corresponded to the overall UCC population, with 81% being men and with a median age of 68 years (range 35–82), as listed in Table 1 [29]. One patient was hospitalization due to a cerebral stroke on the same day as completing informed consent, leaving 78 patients for further QoL and PRO analysis. Twenty-six patients underwent neoadjuvant treatment, while the remaining 53 patients were treated for metastatic disease. Forty-three of the 79 patients (54%) did not complete all planned treatment cycles. The reasons for discontinuation and admission to hospital are shown in Table 1.

Noteworthily, 39% (*N* = 17) of the patients who discontinued treatment did so due to death or intolerable symptoms while 42% (*N* = 18) discontinued treatment due to progression and 19% (*N* = 8) discontinued treatment due to nephrotoxicity. When patients discontinued due to intolerable symptoms, the decision was made by the patient and/or the oncologist based on the severity of symptoms suggesting that continuation was unfavorable. The median number of completed series of chemo- or immunotherapy was four for both the neoadjuvant and metastatic groups. The reasons for hospitalization were primarily (53%) due to actionable symptoms and not related to nephrotoxicity, hematologic toxicity, or thrombo-embolic events, which account for 6%, 26%, and 13%, respectively.

### 3.3. Questionnaire Completion Rate

Of the 78 patients who completed the questionnaires, 21 (27%) completed weekly questionnaires on paper and 57 (73%) patients reported through an electronic platform weekly using the Internet. Seventy-one patients completed the baseline questionnaires (91%). As patients only completed questionnaires while in treatment, a total of 930 questionnaires were sent out, corresponding to 78 patients completing a total of 310 treatment cycles (3 weeks/cycle) with weekly questionnaires. In total, 711 of the 930 questionnaires (76%) were completed. The mean number of questionnaires completed by the patients was 9 (range: 0–20); 90% of the patients completed more than one assessment. The completeness of data over time is shown in Figure 1. The overall completeness of data was 93% (7% missing values). See a detailed description of the adherence data in a separate publication [19].

### 3.4. Health-Related Quality of Life

The mean global QoL was 61 (± SD 24) at baseline and differed between the neoadjuvant (73, ±SD 19) and metastatic group (54, ±SD 24). The difference between the two groups, however, did not persist during treatment (see Table 2), with the global QoL for the metastatic population reaching a mean score of 69 (±SD 21) after three cycles of treatment vs. 72 (±SD 23) for the neoadjuvant population after four cycles of treatment.

Figure 2 graphically displays the development of global QoL and its prevalent subdomains from the QLQ-C30 and the QLQ-BLM30 listed in Table 3 throughout treatment for both the neoadjuvant and metastatic populations: fatigue, pain, and urinary symptoms.

An analysis of mean QoL for the metastatic population was carried out after three cycles of treatment as 24 patients terminated treatment and questionnaire completion after three cycles, with the majority due to progression (*n* = 13), thereby minimizing the false weight of responses from responders to treatment. No statistically significant difference in global QoL at baseline (mean difference −2.4, 95% CI: −17.5–12.6, *p* = 0.75) or after three cycles of treatment (mean difference −2.9, 95% CI: −23.4–17.6, *p* = 0.77) between the two treatment modalities was found.

Patients receiving immunotherapy experienced a statistically significant increase in QoL after three cycles of treatment, although the increase was not clinically important (point estimate: 1.6, 95% CI: 0.3–2.9, *p* = 0.02) [30], as shown in Table 4.

For responders to treatment who continued treatment after three cycles, both groups experienced a statistically significant but not clinically important increase in QoL (chemotherapy estimate: 0.9, 95% CI: 0.4–1.4, *p* = 0.001, immunotherapy estimate: 1.2, 95% CI: 0.6–1.8, *p* < 0.001) [30].

For patients experiencing hospitalization during treatment, the mean global QoL was lower throughout treatment (mean global QoL 60, ±SD 21) compared to the patients who were not hospitalized (mean global QoL 70, ±SD 20), with a mean difference of 8.8, 95% CI 5.8–11.8, *p* < 0.0001. Likewise, statistically significant differences were observed between patients completing treatment vs. those who did not complete treatment (mean global QoL 70 ± SD18 vs. 58 ±SD 23, mean difference = 11.4, 95% CI: 8.3–14.5, *p* < 0.0001). Noteworthily, 23 patients (30%) experienced a decrease in global QoL > 10 points over the course of treatment. No statistically significant differences in characteristics were found between the group of patients who experienced the decrease in QoL vs. those who did not (data not shown).

For the neoadjuvant population, urinary symptoms decreased markedly during treatment. As listed in Table 2, we observed a small increase in fatigue in the neoadjuvant population and a noteworthy decrease in pain among the metastatic patients. For ethical reasons, patients terminating treatment were not required to continue completing questionnaires. Therefore, assessments of the development of QoL and its subdomains after 3–4 cycles represent only that of responders.

### 3.5. PRO-CTCAE Symptoms

At baseline, 44/71 patients (62%) experienced a PRO-CTCAE symptom grade ≥ 2. After three cycles of treatment, 19 of the remaining 38 patients (50%) experienced a PRO-CTCAE symptom grade ≥ 2. Overall, seven patients did not experience a symptom grade ≥ 2. All symptoms grade ≥ 2 over the course of treatment are listed in Table 3, with frequent urination, fatigue, pain, dry mouth, swelling of the arms or legs, abdominal pain, decreased appetite, insomnia, shortness of breath, and nausea being the 10 most frequent symptoms across the entire course of treatment. Decreased libido was not very prevalent overall (14%); however, of the 654 responses to this item, 13% responded that they were not sexually active and 50% answered that they preferred not to respond to this question. The development of symptom burden during the first three cycles of treatment for patients receiving chemotherapy vs. immunotherapy is illustrated in Figure 3 and Table 2 and shows a decrease over time for the patients receiving immunotherapy, while patients receiving chemotherapy experienced stable symptom burden.

When performing the analysis after all cycles of treatment, thereby including responders to treatment, the development was in favor of patients receiving chemotherapy, with a declining symptom burden estimate (−1.0, 95% CI: −1.7–−0.4, *p* = 0.002) compared to that for patients receiving immunotherapy (−0.4, 95% CI: −1.2–0.4, *p* = 0.293).

### 3.6. Anxiety and Depression

Anxiety and depression measured by HADS initially seemed to have a high prevalence in this population as 17 patients (21%) reported a depression score of >11, indicating clinical depression. However, for only 11 of these patients, the high score persisted for more than one cycle of treatment. Likewise, 15 patients scored >11 on the anxiety scale, two of whom persisted with an anxiety score > 11 for more than one cycle of chemotherapy.

## 4. Discussion

In a well-defined group of UCC patients undergoing standard-of-care chemo- or immunotherapy, we observed a stable overall QoL and a concurrent decrease in urinary symptoms over time. In addition, this study demonstrates that a dismal 46% of the patients completed treatment as planned and that 59% of the patients experienced hospital admission during treatment.

Our report of the development of HRQOL in this small population of patients undergoing treatment with chemo- or immunotherapy for metastatic disease compares well with observations based on data from clinical trials although the baseline scores in our study are somewhat lower than that in previous clinical trials [31,32,33]. Our observation of an increase in global QoL seen for the metastatic group receiving chemotherapy is a phenomenon described for responders to therapy in previous studies of metastatic UCC patients [32,34]. Although differences over time are statistically significant, they are small and not clinically meaningful [30]. The decrease in urinary symptoms seems to be in line with the results reported by two clinical trials in metastatic and locally advanced UCC populations, respectively, despite a large difference in both patient population and treatment modality compared to our current setting, and may in part be due to an effect of the treatment resulting in a decrease in symptom burden [32,35]. Additionally, noteworthily, the metastatic group experienced a decrease in fatigue during the first three cycles while the neoadjuvant group experienced a small increase. There may be several possible explanations for this difference. First, the time point of analysis was after three vs. after four weeks of treatment, thus possibly affecting the level of fatigue. Second, differences in prognosis, life expectancy, and thus HRQOL may also impact the individual patient’s feeling of symptom burden. Noticeably, the total PRO-CTCAE symptom burden score for the metastatic population remained stable over time. No previous data of this kind have been reported from clinical trials, yet data of this kind could increase comprehension of oncological treatments and could supplement QoL data when informing patients of upcoming treatments. As outlined in our previously performed systematic review, no literature on the QoL of patients undergoing neoadjuvant chemotherapy exists, making comparisons for this group difficult [8]. However, our data from the neoadjuvant group is comparable to normative QoL scores from the general Danish and European populations completing the EORTC QLQ-C30 (mean overall QoL 71 vs. 73 (Danish) and 66 (European)), despite some differences across gender and age distributions and a substantial disease burden for the patients beginning neoadjuvant treatment [36,37].

In this study, a high percentage of patients experienced hospitalization and a low percentage of patients completed treatment. Previous studies report similar discontinuation rates and median number of cycles to that presented in this study [6,38,39]. Recent data indicate that UCC patients not receiving optimal neoadjuvant chemotherapy are at higher risk of recurrence and death [40]. However, with the limited overall survival of neoadjuvant treatment [38,41,42], one might question the selection of patients eligible for neoadjuvant treatment in this study.

About half of the patients who experienced hospitalization during treatment or discontinued treatment did so due to different degrees of subjective symptoms and not because of objective hematological or nephrological toxicity, with the latter two suspected to not be completely preventable. Furthermore, a high number of patients experienced PRO-CTCAE symptoms grade ≥ 2 throughout treatment. This study therefore highlights an area in which regular symptom reporting and earlier noting of side effects potentially could lead to improved symptom control and thus adherence to treatment. This could prove especially important for the neoadjuvant population as these patients need to maintain a suitable performance status before upcoming surgery [6]. Serious symptoms and associated declines in QoL could lead to earlier cystectomy, when surgically possible.

Severe symptoms, e.g., PRO-CTCAE symptoms ≥ 2, are assumed to affect QoL [43,44]. However, in this study, our data indicate that global QoL remains stable throughout treatment regardless of the treatment modality despite severe symptoms, thereby questioning which symptoms have an impact on QoL for this patient group. Further research into this area is warranted.

Finally, the prevalence of persistent depression seen in this sample population is comparable to that of the general European population in a recent study [45], although 21% of the patients in our study at a single time point all scored as having clinical depression using HADS, as was the case with anxiety for 19% of patients. However, only a few of these patients had persisting symptoms. One explanation for this could be that the clinical staff resolved these issues timely for the patients scoring high on HADS. As such, these numbers emphasize the need for PROs in daily clinical practice to actively monitor and handle these issues before they become persistent.

Planned as a descriptive, prospective study of a group of previously overlooked patients, the number of patients in this study represents a clear limitation that makes strong conclusions impossible; hence, wide standard deviations are shown in Figure 2. Caution should especially be upheld when interpreting the development of overall HRQOL and the subdomains listed, and the analysis hereof can thus only be regarded as a guidance. Importantly, the HRQOL of patients receiving neoadjuvant vs. metastatic treatment may differ markedly, as may the HRQOL of metastatic patients receiving 1st vs. 2nd or 3rd line treatments. We included patients in all lines of treatment, and the data presented may therefore not be directly representative of the metastatic population as a whole. Additionally, the lack of information on previous surgery and types of urinary diversions for especially the metastatic population represents a limitation, and the results presented should thus be interpreted with care.

Additionally, as the completion rate of the weekly questionnaires falls during treatment, the reliability of the information is, at best, limited. Specifically, missing responses to the questions concerning sexual symptoms give rise to uncertainties about the interpretation of the overall QoL, as the sexual domain is known to impact overall QoL [46]. The missing responses also suggest that the taboo surrounding talking about sexuality still dominates and shed lights on an area in need of extended efforts for this group of patients. The reasons for not completing the questionnaires were not required from the participants due to ethical considerations. The completion rate is, however, at a level observed in similar studies, even though this is an elderly and often comorbid population [12,47,48]. Finally, as PRO completion was conducted on both paper and electronically but only electronic completions were conveyed to the treating physician, this may have affected the individual patient’s symptom burden and, hence, HRQOL through earlier symptom management by ePROs.

## 5. Conclusions

This study is the first to present quantitative HRQOL data from UCC patients receiving neoadjuvant chemotherapy and from UCC patients with metastatic disease in standard-of-care chemo- or immunotherapy. Though the patient numbers are limited, the data point to a need for extended efforts in this patient group to improve the clinical outcomes, which in this study were found to be rather dreary. Based on these PRO data, we hypothesize that better supportive care in terms of regular symptom reporting with PROs followed by targeted clinical interventions could become a method to improve overall outcomes for these patients regarding hospital admissions, adherence to treatment, and QoL. We therefore applied this knowledge to a randomized trial (NCT03584659) conducted in this comorbid and previously neglected patient group, and we await the results, to be consolidated in 2021.

## Figures and Tables

**Figure 1 jcm-10-01852-f001:**
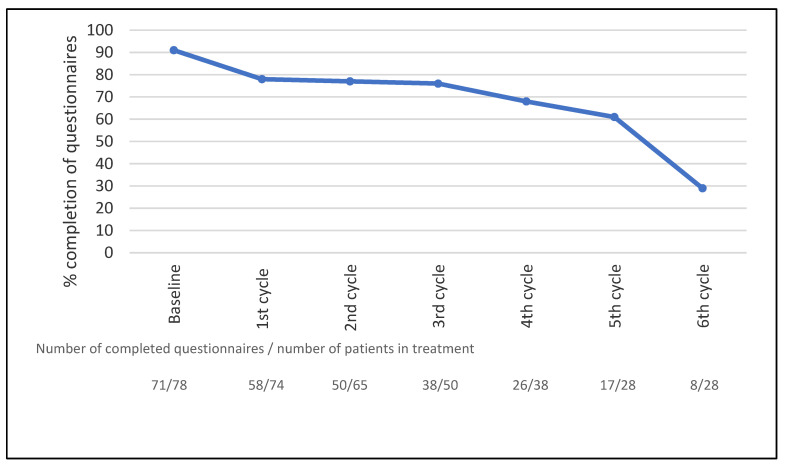
Completeness of data during treatment, *n* = 78. The curve shows the percentage of patients in active treatment completing the questionnaires. The treatment cycle interval was three weeks for all treatment regimes.

**Figure 2 jcm-10-01852-f002:**
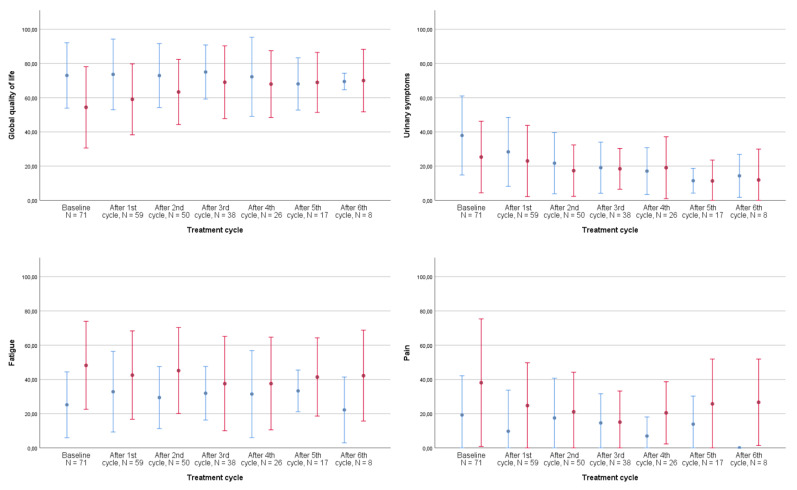
Development of quality of life during treatment by disease stage, all patients, *N* = 78. The shown development of quality of life and its subdomains are the computed scores from the EORTC QLQ-C30 core module and QLQ-BLM30 muscle-invasive bladder cancer-specific module from the most frequent symptoms in Table 2. Number of patients completing questionnaires: baseline: 71, after 1st cycle: 59, after 2nd cycle: 50, after 3rd cycle: 38, after 4th cycle: 26, after 5th cycle: 17, and after 6th cycle: 8. Error bars represent ±1 standard deviation (SD). Red bars: metastatic patients, blue bars: neoadjuvant patients.

**Figure 3 jcm-10-01852-f003:**
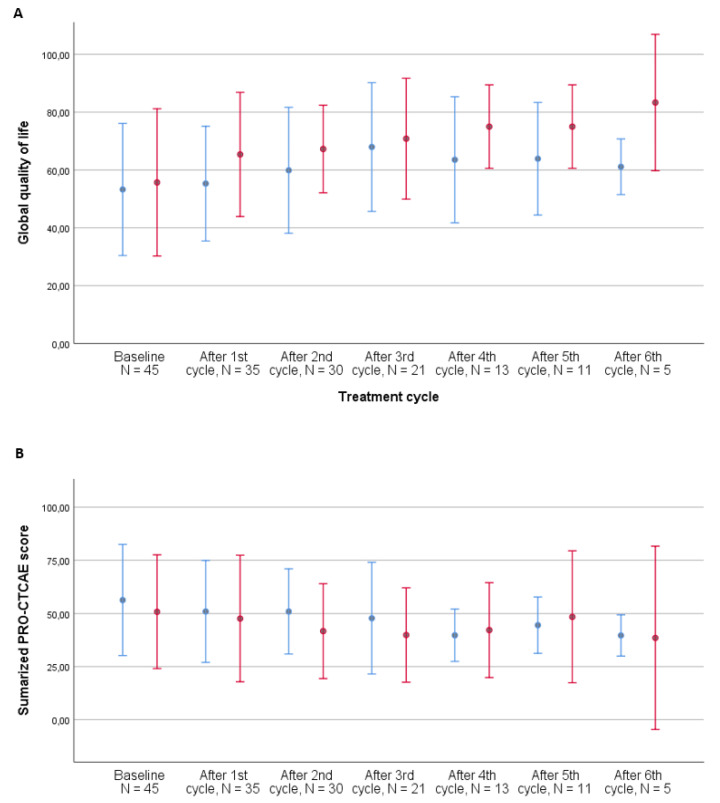
Development of global quality of life (**A**) and summarized PRO-CTCAE score (**B**) during treatment by treatment type, metastatic patients, *N* = 53. Number of patients completing questionnaires: baseline: 45, after 1st cycle: 35, after 2nd cycle: 30, after 3rd cycle: 21, after 4th cycle: 13, after 5th cycle: 11, and after 6th cycle: 5. PRO-CTCAE questionnaires with less than 50% completed items were excluded from the analysis. Error bars represent ±1 standard deviation (SD). Blue bars: chemotherapy. Red bars: immunotherapy.

**Table 1 jcm-10-01852-t001:** Patient characteristics, *N* = 79.

Clinical Data	Total *N* = 79 (%)	Neoadjuvant Group *N* = 26 (%)	Metastatic Group *N* = 53 (%)
Gender			
Men	64 (81)	21 (81)	43 (81)
Women	15 (19)	5 (19)	10 (19)
Median age, years (range)	68 (35–82)	66 (52–77)	68 (35–82)
Disease			
Neoadjuvant	26 (33)	-	-
Metastatic	53 (67)	-	-
Treatment *			
Cisplatin-gemcitabine	46 (59)	25 (96)	21 (40)
Carboplatin-gemcitabine	9 (12)	1 (4)	8 (15)
Vinflunine	2 (3)	0 (0)	2 (4)
Vinflunine-Gemcitabine	1 (1)	0 (0)	1 (2)
Pembrolizumab	20 (26)	0 (0)	20 (39)
Treatment completion			
No	43 (54)	11 (42)	32 (60)
Yes	36 (46)	15 (58)	21 (40)
Reason for discontinuation			
Progression	18 (42)	3 (27)	15 (47)
Nephrotoxicity	8 (19)	4 (36)	4 (13)
Death	4 (9)	0 (0)	1 (3)
Hematological toxicity	2 (5)	0 (0)	2 (6)
Declining performance status	2 (5)	1 (9)	1 (3)
Infection	2 (5)	0 (0)	2 (6)
Dyspnea	1 (2)	1 (9)	0 (0)
Constipation	1 (2)	1 (9)	0 (0)
Colitis	1 (2)	0 (0)	1 (3)
Neuropathy	1 (2)	0 (0)	1 (3)
Otologic toxicity	1 (2)	1 (9)	0 (0)
Intolerable decrease in QoL	1 (2)	0 (0)	1 (3)
Other	1 (2)	0 (0)	1 (3)
Admission to hospital			
No	32 (41)	15 (58)	17 (32)
Yes	47 (59)	11 (42)	36 (68)
Reason for hospital admission			
Hematological toxicity	12 (26)	3 (27)	9 (25)
Infection	11 (23)	3 (27)	8 (22)
Thrombo-embolic events	6 (13)	1 (9)	5 (14)
Pain	4 (9)	0 (0)	4 (11)
Nephrological toxicity	3 (6)	1 (9)	2 (6)
Dyspnea	2 (4)	0 (0)	2 (6)
Nausea/vomiting	2 (4)	1 (9)	1 (3)
Constipation	1 (2)	1 (9)	0 (0)
Pneumonitis	1 (2)	0 (0)	1 (3)
Nephritis	1 (2)	0 (0)	1 (3)
Dehydration	1 (2)	1 (9)	0 (0)
Declining performance status	1 (2)	0 (0)	1 (3)
Ileus	1 (2)	0 (0)	1 (3)
Other	1 (2)	0 (0)	1 (3)

* Due to one patient experiencing a cerebral stroke shortly after enrollment into the study, the data are only shown for 78 patients.

**Table 2 jcm-10-01852-t002:** Development of global quality of life and symptom scores during treatment, *N* = 78.

**Neoadjuvant Group**								
	Baseline				After 4 Cycles			
	Mean	Min	Max	SD *	Mean	Min	Max	SD *
Global Quality of Life	73	25	100	19	72	25	100	23
Pain **	19	0	100	23	7	0	33	11
Fatigue **	25	0	67	19	31	0	78	25
Urinary symptoms **	38	0	76	23	17	0	38	14
**Metastatic Group**								
	Baseline				After 3 cycles ***			
	Mean	Min	Max	SD *	Mean	Min	Max	SD *
Global Quality of Life	54	0	100	24	69	33	100	21
Pain **	38	0	100	37	15	0	50	18
Fatigue **	48	0	100	26	38	0	89	28
Urinary symptoms **	25	0	76	21	18	5	43	12

Global quality of life, pain, and fatigue scores are from QLQ-C30, and urinary symptoms are from QLQ-BLM30. * SD: standard deviation. ** Note that, for the symptom scores, the scale is reversed as compared to the global QoL score, with 0 indicating no symptom burden and 100 indicating severe symptom burden. *** Analysis was carried out after 3 cycles of treatment because many metastatic patients did not continue past three cycles due to progression to avoid underestimating the symptom burden by only including responders to treatment.

**Table 3 jcm-10-01852-t003:** Prevalence of PRO-CTCAE symptoms grade ≥ 2 during treatment, *N* = 78.

Symptom	Total	Baseline	After 1st Cycle	After 2nd Cycle	After 3rd Cycle	After 4th Cycle	After 5th Cycle	After 6th Cycle
	%	%	%	%	%	%	%	%
Reports, *N*	449	71	59	50	38	26	17	8
Frequent urination	37	45	35	38	35	36	31	13
Fatigue	35	34	37	38	26	27	59	28
General pain	31	45	42	27	26	15	35	13
Dry mouth	23	23	28	20	26	23	18	13
Swelling of arms or legs	23	21	18	25	24	31	41	0
Abdominal pain	21	27	19	22	21	19	18	0
Decreased appetite	21	23	19	20	19	24	24	0
Insomnia	20	34	19	17	13	19	6	0
Shortness of breath	20	13	24	29	18	19	18	0
Nausea	19	14	16	22	24	16	24	25
Urinary urgency	18	40	27	22	19	20	13	13
Muscle pain	18	19	21	16	16	12	18	13
Constipation	15	14	21	12	16	15	12	0
Sad	15	21	16	16	11	8	6	13
Decreased libido	14	27	8	9	8	16	6	13
Joint pain	13	15	14	12	8	12	18	25
Taste changes	13	8	10	16	16	19	18	13
Heart palpitations	12	14	12	16	8	4	6	13
Dry skin	12	13	10	10	13	12	12	13
Numbness & tingling	11	13	3	8	11	12	30	25
Painful urination	11	22	7	8	11	8	0	0
Diarrhea	9	14	7	8	5	12	0	13
Urinary incontinence	9	16	9	7	5	8	6	0
Anxiety	9	19	9	2	5	8	0	0
Discouraged	9	8	9	13	8	4	12	0
Headache	8	11	8	6	3	8	12	13
Ringing in ears	8	3	7	6	11	15	12	13
Cough	7	5	11	6	5	12	6	0
Dizziness	7	5	5	8	11	4	6	13
Increased sweating	6	13	5	4	8	0	0	0
Heartburn	6	6	12	6	3	4	0	0
Hair loss	6	0	9	6	8	4	12	13
Itching	6	8	5	4	3	8	6	0
Chills	6	6	9	2	3	12	0	0
Difficulty swallowing	5	6	5	2	8	0	12	0
Hot flashes/flushes	4	10	5	0	3	0	0	0
Concentration	3	6	0	4	3	4	0	0
Mouth/throat sores	3	3	2	4	3	0	6	0
Memory	2	3	0	4	3	4	0	0
Blurred vision	2	2	2	0	3	4	0	0
Vomiting	2	2	0	2	3	0	0	13
Rash§ (yes/no)	65	65	64	63	68	62	77	63
Change in usual urine color § (yes/no)	24	27	19	22	26	27	24	38
Pain/swelling at injection site § (yes/no)	24	11	23	20	40	39	35	13
Hives § (yes/no)	8	5	7	12	11	12	0	13

All symptoms included in the table are represented by the initial PRO-CTCAE item according to the symptom. If the symptom is not represented by a frequency item, the following (often severity item) is displayed. § These questions were ‘Presence’ (yes/no) questions. The percentages relate to the number of patients experiencing the symptom (no grading).

**Table 4 jcm-10-01852-t004:** Development of global quality of life and summarized PRO-CTCAE scores during three cycles of treatment for metastatic disease, *N* = 53.

		Confidence Interval		*p*-Value
	Estimate	Lower	Upper	
Global quality of life				
Chemotherapy	1.07	−0.07	2.21	0.066
Immunotherapy	1.62	0.26	2.97	0.02
Summarized PRO-CTCAE score				
Chemotherapy	−0.82	−0.23	0.69	0.286
Immunotherapy	−1.89	−3.42	−0.16	0.032

Analyses were carried out for all metastatic patients from baseline to after the 3rd cycle of treatment. PRO-CTCAE questionnaires with less than 50% completed items were excluded from the analysis.

## Data Availability

The data presented in this study are available upon request from the corresponding author. The data are not publicly available due to the General Data Protection Regulation.

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
