# Peer review of "Patient-Reported Outcomes, Health-Related Quality of Life, and Clinical Outcomes for Urothelial Cancer Patients Receiving Chemo- or Immunotherapy: A Real-Life Experience"

_jcm, 2021, doi:10.3390/jcm10091852_

Round 1

Reviewer 1 Report

Abstract:

  • Local muscle invasive- do they mean locally advanced or localized bladder cancer?
  • Is this study met RCC including upper tract as well?
  • UCC given the impression that other variants (adenocarcinoma, SCC, etc) were not included.

Manuscript:

  • This study looks more into the effects of chemotherapy. In the abstract, it says “treatment” not specifying chemotherapy. Treatment would include surgery, immunotherapy, etc. Please make it clear that it is only for chemotherapy.
  • Consider re=phrasing “It therefore seems reasonable to register QoL during treatment for 40 these patients, albeit no QoL data from UCC patients receiving standard chemotherapy 41 as neoadjuvant treatment or for metastatic disease exists [8]”.
  • What do the authors mean with “However, it is well 43 known, that such data may not be representative of patients with UCC receiving standard treatment [9,10]”?
  • Do you give immunotherapy as neoadjuvant therapy outside clinical trial? If yes, what is the reference? If not, please consider revising “Initiating chemo- or immunotherapy as standard therapy (outside clinical trials) as neoadjuvant or metastatic treatment”.
  • At what time points after/during treatment were these surveys completed? Was there a baseline before starting? It seems from the results that there were baseline questionnaires and that these were completed weekly. This needs to be clearly mentioned in the methods.
  • Sample size is very small
  • Combining neoadjuvant with metastatic treatment may not be the best approach. These patient s have totally different prognosis. I’d separate these 2 or even compare them. Age, CCI, disease status, response to treatment, all of these will be different.

Reviewer 2 Report

it should be UC and not UCC- urothelial cancer,

In Results- clinical data- it looks incomplete- 81% men but how many women? and medain age 68 years (range)

difficult to understand- that 54% didnt complete treatment and this didnt reflect on the QOL?

the small number of patients makes it a irrelevant exercise

Round 2

Reviewer 1 Report

No more concerns.

Reviewer 2 Report

ok for publication